# A Method for Detecting Pathologies in Concrete Structures Using Deep Neural Networks

**Joel de Conceição Nogueira Diniz** [1], **Anselmo Cardoso de Paiva** [1], **Geraldo Braz Junior** [1], **João Dallyson Sousa de Almeida** [1], **Aristofanes Correa Silva** [1], **António Manuel Trigueiros da Silva Cunha** [2,3] and **Sandra Cristina Alves Pereira da Silva Cunha** [2,4,*]

1. UFMA/Computer Science Department, Universidade Federal do Maranhão, Campus do Bacanga, São Luís 65085-580, Brazil; joel.diniz@discente.ufma.br (J.d.C.N.D.); paiva@nca.ufma.br (A.C.d.P.); geraldo@nca.ufma.br (G.B.J.); jdallyson@nca.ufma.br (J.D.S.d.A.); ari@nca.ufma.br (A.C.S.)
2. UTAD/Engineering Department, Universidade de Trás-os-Montes e Alto Douro, 5000-801 Vila Real, Portugal; acunha@utad.pt
3. INESC-TEC—Institute for Systems and Computer Engineering, Technology and Science, 4200-465 Porto, Portugal
4. CMADE—Centre of Materials and Building Technologies, UTAD, 5000-801 Vila Real, Portugal
* Correspondence: spereira@utad.pt

**Abstract:** Pathologies in concrete structures, such as cracks, splintering, efflorescence, corrosion spots, and exposed steel bars, can be visually evidenced on the concrete surface. This paper proposes a method for automatically detecting these pathologies from images of the concrete structure. The proposed method uses deep neural networks to detect pathologies in these images. This method results in time savings and error reduction. The paper presents results in detecting the pathologies from wide-angle images containing the overall structure and also for the specific pathology identification task for cropped images of the region of the pathology. Identifying pathologies in cropped images, the classification task could be performed with 99.4% accuracy using cross-validation and classifying cracks. Wide images containing no, one, or several pathologies in the same image, the case of pathology detection, could be analyzed with the YOLO network to identify five pathology classes. The results for detection with YOLO were measured with mAP, mean Average Precision, for five classes of concrete pathology, reaching 11.80% for fissure, 19.22% for fragmentation, 5.62% for efflorescence, 27.24% for exposed bar, and 24.44% for corrosion. Pathology identification in concrete photos can be optimized using deep learning.

**Keywords:** concrete pathologies; deep learning; classification; detection





## 1. Introduction

Civil Construction is of enormous relevance for its relationship with the development of urban centers and for representing an important sector for the economy, generating investments and jobs [1]. The maintenance of buildings is fundamental because of their direct relationship with people's safety. Inadequate, or even non-existent, maintenance can result in the loss of a building or, even worse, in accidents to users, which can even result in severe and fatal accidents.

Reinforced concrete is the world's second most used building material, providing a versatile, durable, affordable, functional, and attractive construction [2]. Its high compressive strength property allows this material to resist this very present stress in structures, and its deficiency in tensile strength can be compensated with its association with steel [1]. The possibility of easily molding concrete, even on the site where it will be used, allows a versatility that also provides a desirable advantage that places it as one of the main alternatives in building construction. It is a very durable building material but can deteriorate due to exposure to severe conditions associated with the environment, loading, effects of

aggressive actions, embedded metal corrosion, frost overloading, concrete resistance to volume changes, abrasion/erosion, and chemical actions.

This maintenance is essential for reinforced concrete structures, given their wide use and the collapse impacts of these structures.

Although concrete has enormous advantages, it is subject to pathologies that can compromise its strength. The great danger is that concrete structures lose the calculated resistance for which they were designed. For this reason, it must be monitored to prevent the onset or worsening of pathologies that may develop throughout life. There are several techniques to monitor concrete structures and identify their pathologies, such as instrumentation, image analysis, and on-site inspection, which is very usual for inspecting its surface because most of the concrete pathologies end up representing specific characteristics in its surface [3].

Several pathologies can be analyzed via images of concrete structures since their effects can appear on the surface [3]. This is, for example, the case of fissures or cracks, fragmentation of part of the concrete, efflorescence, corrosion stains, and exposed steel bars. These pathologies are explained along with image presentation in the images section.

Cracks represent one of the main concerns with concrete. This paper works on the classification with cropped images of the pathology of cracks. The largest availability of datasets is related to this pathology exactly because of its importance. These justifications result in a greater need for understanding this pathology.

The cracks can be observed according to the form of the manifest in "geometric or isolated" or "mapped or disseminated". According to their activity, they can be "active or alive" or "passive". The cracks can also be observed according to the variation of their opening in "seasonal" or "progressive".

The way cracks appear varies significantly, but observing the manifestation is fundamental for a study that leads to understanding this pathology. The design formed by the cracks results in the first geometric or disseminated definition. This first observation can be subdivided into active or passive, depending on whether the activity is progressing or has stabilized. Finally, concerning the time that the crack varies its opening or closing, around an average value, they can be observed as seasonal or progressive. The causes of these variations need to be investigated; for example, the opening can oscillate due to temperature or humidity variation.

Manual visual inspection is one of the primary methods for determining a wired concrete structure's physical and functional condition. This inspection should be performed regularly to ensure the structure continues functioning as intended. However, many incidents continue due to inadequate inspection and condition assessment.

Visual inspection of surfaces is one of the primary methods for determining a reinforced concrete structure's physical and functional condition. This inspection should be performed regularly to ensure the structure continues functioning as intended. However, many incidents continue due to inadequate inspection and condition assessment This visual inspection is very time-consuming and poses a safety risk to inspectors. It is a method that is subjective to the skills, expertise, and experience of the engineers performing the inspection. Automated inspection methods have been developed using image acquisition and machine learning algorithms to overcome these limitations. This automation can result in a more cost-effective and efficient method and reduces the risk of workplace accidents. In addition, it can be objective and more reliable since the condition is defined by an algorithm being reproducible. Many studies have investigated the possibility of automated inspection of concrete structures using deep learning methods.

The literature presents important works on concrete pathologies and artificial intelligence. Cascaded deep neural networks automatically detect image damage and cracks [4]. Research has explored how to improve crack detection and recognition based on Convolutional Neural Networks [5]. An overview of the challenges associated with automatic concrete crack detection in the presence of shadows is presented in [6]. The paper explains how to increase the robustness of material-specific deep learning models for crack detection

in different materials [7]. Automated crack detection and analysis based on artificial neural networks has been used to inspect concrete structures [8]. Works have presented crack detection in concrete images using machine learning classification techniques [9].

Deep and convolutional neural networks have been compared in terms of image-based concrete surface crack detection [10]. Autonomous structural visual inspection has been presented using region-based deep learning to detect various types of damage [11]. Deep learning-based crack damage detection using convolutional neural networks is presented in [12]. Additionally, concrete crack detection based on deep learning image classification mentioning inspection when combined with unmanned aerial vehicles (UAV) is applied in [13]. These papers support pathology studies using artificial intelligence.

Other papers in this line of research present relevant studies on using machine learning and concrete. The interest in monitoring the health of the existing bridge heritage has been combined with automation techniques using machine learning [14]. Vulnerability analysis of existing buildings indicating seismic vulnerability by exploiting available photographs with machine learning techniques is the topic of [15]. Quickly assessing the spatial distribution and severity of building damage for post-event emergency response and recovery using machine learning is achieved in [16].

This paper proposes two methods to detect and classify pathologies in images of concrete structures using convolutional neural networks. We investigated two possible approaches to deal with the reinforced concrete pathologies' automatic visual inspection. The first approach uses cropped images with a detected pathology to identify its type (classification). The other possibility is detecting in a reinforced concrete surface image of the specific regions where there are none or several pathologies (detection).

The paper presents other work using classification with results close to those experienced. This paper proposes integrating classification dedicated to focused pathology images with detection in wide images of the structures.

These methods contribute to the analysis of concrete pathology images by saving time in image analysis with the use of computer vision, reducing errors, which are specifically due to human errors originating from the need to repeat the tedious manual operation of a large volume of data; providing operation with a massive amount of image data; and making use of datasets present in the literature, which can stimulate the emergence of new resources.

## 2. Materials and Methods

This paper proposes two methods, one for detecting and the other to classify pathologies in reinforced concrete structures using deep neural networks. Figure 1 summarizes a diagram of this method.

The proposed method can also be defined by the main steps that constitute it, which are the following:

- Image acquisition;
- Image preprocessing;
- Pathology detection;
- Pathology classification.

The initial step is the acquisition of the images dataset for model induction and, after this, to perform the detection or classification task. In this work, we used only publicly available image datasets. After the acquisition, the proposed methods use acquired images of the structure to be analyzed. The acquired images are generally not suited for detection or classification tasks. Thus, we propose the use of pre-processing steps to improve their quality. After this, the proposed method can detect from the wide language dataset where the pathologies are and what is done in the pathology detection step. The detected regions of the wide-angle images may now be processed by the pathology classification step to determine to which classes they belong.

The following sections detail the steps of the method.

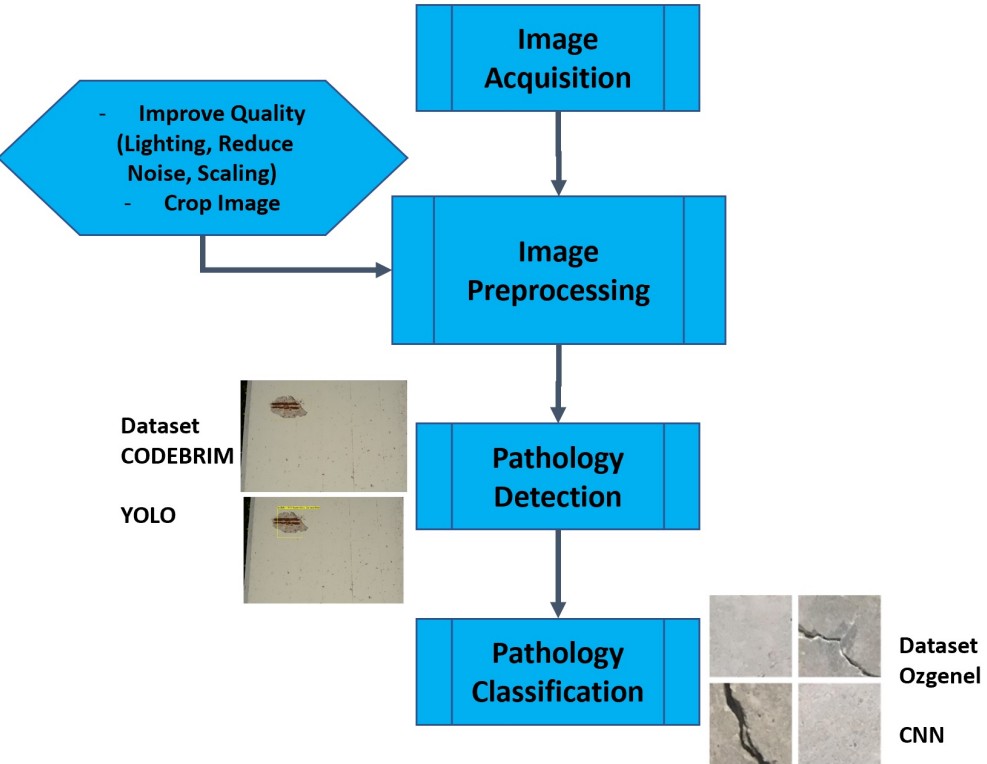

**Figure 1.** Proposed method as diagram.

## 2.1. Image Acquisition

This paper deals with five types of concrete pathologies, cracking, steel corrosion spots in concrete, concrete splintering, efflorescence, and exposed steel bars in reinforced concrete. Determining the pathologies to be studied leads to choosing appropriate datasets that enable detection and classification for these pathologies.

The CODEBRIM [17] dataset is a base annotated to enable the detection of pathologies in concrete structures. This dataset allows the detection of five pathologies, cracking, exposed steel bar, concrete fragmentation, efflorescence on the concrete surface, and corrosion spots on the concrete surface.

The "Concrete Crack Images for Classification" dataset, Özgenel [18–20], fits the case of cropped image analysis for a specific pathology, in this case, cracks in concrete. This dataset contains 20,000 images for each classification, for a total of 40,000 images of the types. The images are in 227 × 227 pixel format with RGB channels.

One of the concrete pathologies that cause a lot of concern is the presence of cracks or fissures. This pathology has its severity evaluated according to the thickness of its opening. Figure 2A shows a crack case, a pathology that is annotated in both [17,18] datasets. Figure 2B shows a cropped image of a case of an exposed steel bar. This is a case of pathology in reinforced concrete, a scenario where you have concrete and steel as structural elements. Figure 2C shows part of an image of a fragmented concrete. The fragmentation can start in another pathology, such as cracking or splitting. It can be a consequence of the expansion of steel that suffers corrosion or even loss of the structure that has suffered some impact. Figure 2D shows an image cutout with a case of efflorescence, which is white spots on the surface of concrete produced by the accumulation of crystalline deposits. In the last case analyzed, Figure 2E, we present an example of a corrosion spot. Steel corrosion occurs in the case of reinforced concrete, concrete, and steel, collaborating to ensure the strength of the structure.

**Pathologies in Concrete**

(C) Fragmentation

(A) Crack

(D) Efflorescence

(B) Exposed Bar

(E) Corrosion spot

**Figure 2.** Pathologies in Concrete [17].

*2.2. Image Preprocessing*

The quality of the acquired images is a fundamental factor for the correct detection and classification of possible pathologies present in the concrete structure to be evaluated. Poor-quality images may result in non-detection or even misclassification of the detected pathology. Possible noise needs to be removed from the images.

Although every care is required for good image acquisition, the conditions may not be favorable [21], resulting in images that need to be enhanced for analysis. Some techniques can be employed for image enhancement, such as correction of illumination, sharpening, removing noise, and correcting any imperfections the image may have.

The method proposed in this paper suggests image enhancement so that the poor quality of an image does not interfere with the pathology identification result. Still, the experiments did not use these techniques since the images were from public datasets with images already ready for pathology identification processing. Several techniques can be employed for image enhancement; the choice depends on the problem the image contains, which is related to the conditions under which the image will be taken, as well as the conditions of the environment where the structure to be analyzed is located.

One possibility is the use of filters for image enhancement. It is possible to soften the image with a filter that works with the transition of intensities in the image. Enhancement filters can increase the sharpness of the image, emphasizing edges and details of the image, and can work not only on sharpness but also on contrast and detail enhancement. Image intensity adjustment can also be worked on to improve brightness and contrast to make the image sharper and more detailed.

Filters can also be used to remove or reduce noise. The filter to be used will depend on the type of noise in the image, and tests need to be performed to see whether the processing results in the desired image. Color correction can be worked on to correct the distribution of colors in the image. This technique can improve image quality and result in a more realistic image with vibrant colors.

It is important to note that each image produced is a unique artifact, depending on many variants, the way it was made, the equipment used, and the environmental conditions at the time it was made, among many others. For this reason, therefore, each case needs to be analyzed and a strategy adopted for the need of each specific case.

*2.3. Pathology Detection*

Once the images are acquired, the first step is to determine if any pathology may be visually detected and determine the area of the image corresponding to each pathology (bounding box). For this task, we propose to use the one-stage detector YOLOv4 neural

network architecture [22]. YOLO, an abbreviation for "You Only Live Once", is a neural network for real-time detection due to its enormous speed and accuracy for this problem, proposed in [23]. It has evolved through several versions, and we choose version 4, even though there are newer versions.

YOLO essentially redefines object identification as a regression issue and, as it achieves a positive balance of detection accuracy and speed, has become one of the quickest object detection models. The detection pipeline for YOLO separates a picture into regions and forecasts border boxes, probabilities, and conditional class probabilities.

The YOLO architecture is an end-to-end connected network for making bounding box predictions with its probabilities for classes at once, Figure 3. The model uses a single fully connected layer for predictions. YOLO uses a single interaction to propose the regions of interest, while other methods perform multiple interactions for the same image [23]. Initially, the image is split into an S × S grid, Figure 3. The cell located in the object's center will be responsible for detecting the object. Bounding box and confidence scores will be predicted by each cell, the score defining the model's confidence that the bounding box contains the object and with what precision. YOLO uses the concept of Intersection over Union, IoU, to determine the most representative bounding box.

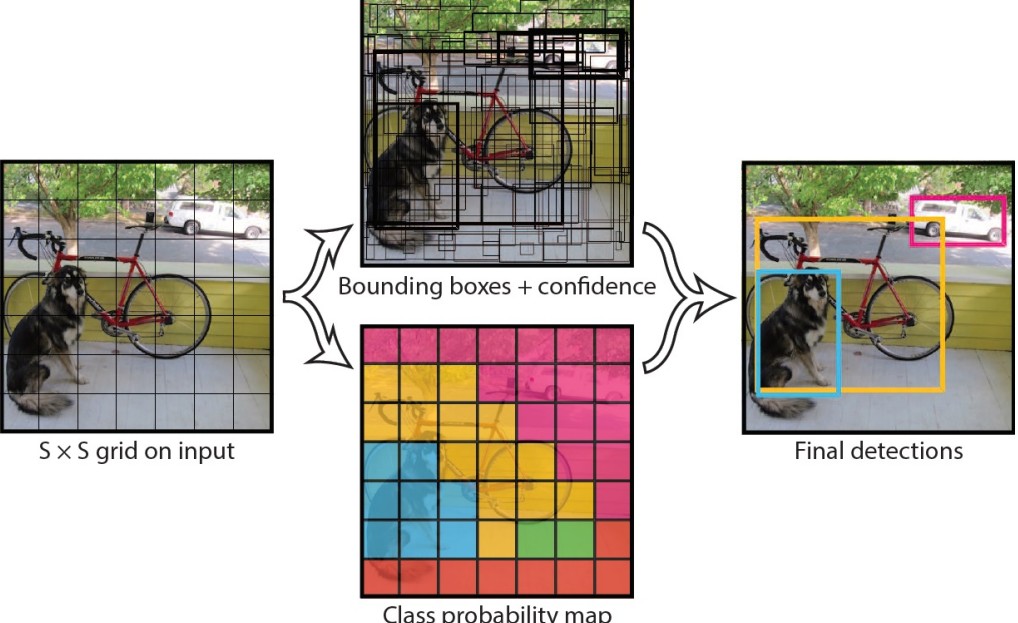

**Figure 3.** YOLO Detection pipeline. The input image was divided into S × S grids. Predicted bounding boxes and confidence and a class probability map for each grid cell are generated. The final detection is generated [23].

YOLOv4 is made up of Cross Stage Partial Darknet-53 (CSPDarknet-53), Spatial Pyramid Pooling (SPP) block, and Path Aggregation Network (PANet). The CSPDarknet-53 is a cutting-edge backbone that can enhance the learning capability of CNNs. To broaden the receptive area and isolate the most important context features, the SSP block is put over CSPDarknet-53. In YOLOv4, PANet is used for parameter aggregation with various detector levels instead of the FPN used in YOLOv3 for object detection. In addition, Mosaic and Self Adversarial Training (SAT) are used to supplement the training pipeline data in YOLOv4.

The detection task also involves classification after the area of possible pathology has been identified. The preparation of the detection deep learning network consists in training the database to determine the learning weights for the network. Once the training is complete, the learning can localize the trained pathologies to any other image, enabling

classification and the location indicated by the bounding box if it contains them. If the pathology area is already defined, classification can be direct in the case of cropped images.

The nets must be trained with the appropriate dataset according to the intended use, classification, or detection. The Ozgenel [18] dataset allows for binary classification, concrete with and without cracks. The dataset CODEBRIM [17] allows the use of detection, in this case, five concrete pathologies.

The algorithm uses a simple CNN to detect objects in the image. Figure 4 shows the CNN model architecture used by YOLO [23].

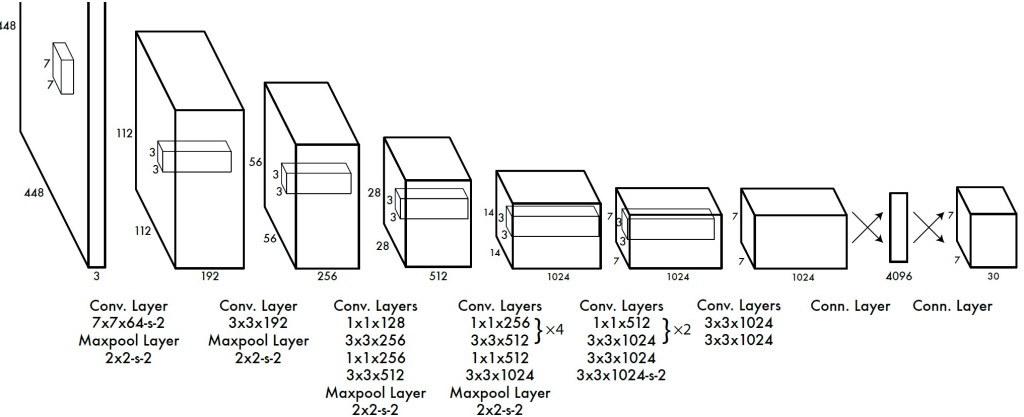

**Figure 4.** YOLO Architecture [23].

The network makes use of transfer learning. The initial twenty layers use this concept, starting from ImageNet pre-training, using a temporary grouping with a fully connected layer. Performance is improved due to the addition of convolution and a pre-trained network. The network ends with a fully connected layer that defines the probabilities and location of the bounding box [23].

It was necessary to adapt the annotation files from the bounding box to use YOLO. The dataset [17] has bounding box annotations relative to the corners of the rectangles, the endpoints of the main diagonal of the rectangle representing the bounding box. YOLO refers to the bounding box in another way, using the coordinate of the center of the rectangle plus the dimensions of the rectangle, height, and width. Therefore, for the experiment, it was necessary to perform a file conversion. The use of an algorithm allows the file to be converted automatically.

The metric used to evaluate the detection result is mPa, mean Average Precision, which considers Precision and Recall, the average of the two values. Precision represents how good the predictions are, and recall represents how good all the positives are.

### 2.4. Pathology Classification

Once the pathologies are detected, the cropped images corresponding to each found pathology are generated. This led to the need for a step to classify these cropped images in the classes associated with the pathology types.

In this work, we propose using a CNN network to work on cropped images for pathology classification. The CNN has been configured for the architecture shown in Figure 5.

Loss is calculated to measure the errors of the classification. The loss represents an essential parameter in neural networks and is calculated as the difference between the achieved and intended results by the deep learning model. Loss functions derive gradients by updating the weights that define neurons' importance and the network's learning. The lower the score, the better the result, with values close to zero representing the best score and values close to 100% representing the worst score.

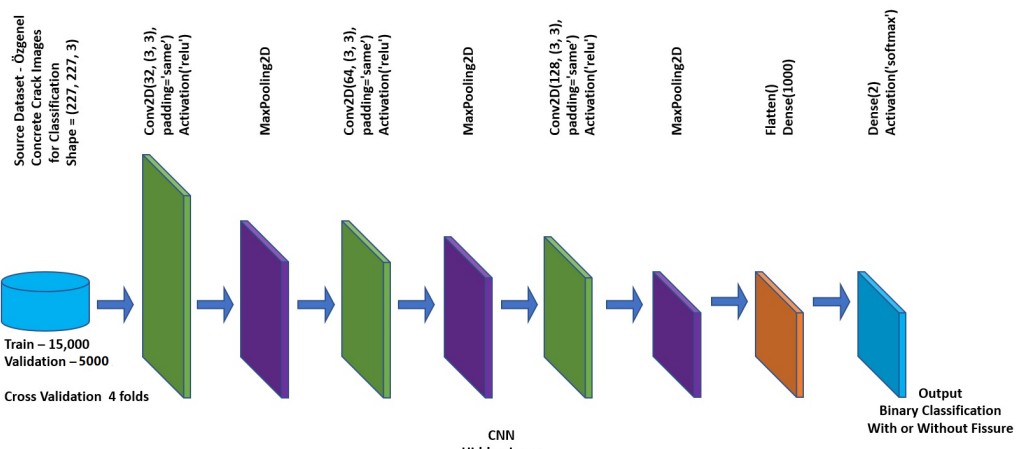

**Figure 5.** CNN Architecture and configuration.

## 3. Experiments and Results

To validate our proposal, we performed two separate experiments, one with a dataset comprising wide-angle concrete structures annotated for detection and one for classification. Thus, the first experiment used the dataset CODEBRIM [17] to evaluate the proposal for pathology detection of five classes of pathologies. The second experiment performs the classification of the fissure pathology in cropped images using the dataset Ozgenel [18].

### 3.1. Detection of Pathologies in Concrete

The detection experiment was done using the dataset [17], applying the YOLO architecture in version 4. It was designed to enable the ability to identify concrete pathologies in wider images, where we do not know if there is a pathology or where the pathologies are present.

The dataset [17] was split into two subsets for training and for testing. A total of 840 images were used for training and 212 for testing, using a total of 1052 images from the dataset. Defect numbers for the following classes: crack—2507, spallation—1898, efflorescence—833, exposed bars—1507, and corrosion stain—1559 [24].

The training was performed for 1500 epochs, and starting at epoch 300, validation tests were performed every 200 epochs with pre-selected images. The metric used was mAP, mean Average Precision; this metric considers precision and recall, and the average of the two values, precision being how good the predictions are and recall being how good all the positives are.

Figure 6 displays the result graphically, and, in this graph, it is better to visualize the training evolution for the five trained classes. All classes have an evolution until epoch 900, but at epoch 1000, efflorescence and cracking become worse, while fragmentation, exposed bar, and corrosion still evolve. At epoch 1100, all classes worsen, resuming their evolution until epoch 1300 and worsening again at epoch 1500. The best performance, therefore, occurred at epoch 1000 for the classes fragmentation, corrosion, and exposed bar. As for cracking and efflorescence, despite the drop in performance after epoch 900, there is a recovery, and the best performance is reached at epoch 1300. Therefore, the results for detection with YOLO were measured with mAP, mean Average Precision, for five classes of concrete pathology, reaching 11.80% for fissure, 19.22% for fragmentation, 5.62% for efflorescence, 27.24% for exposed bar, and 24.44% for corrosion.

Figure 7 shows cases of concrete pathology. It is possible to see exposed steel bars; at this same point, there is concrete fragmentation, and the steel bars show a case of corrosion. This figure was used in the trained detection network to test which pathologies could be identified.

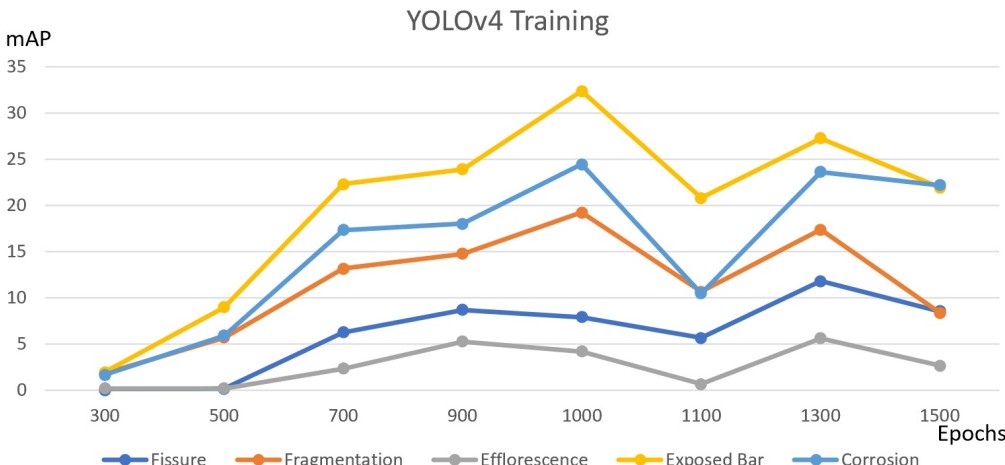

**Figure 6.** Training with YOLOv4 on the CODEBRIM dataset [17].

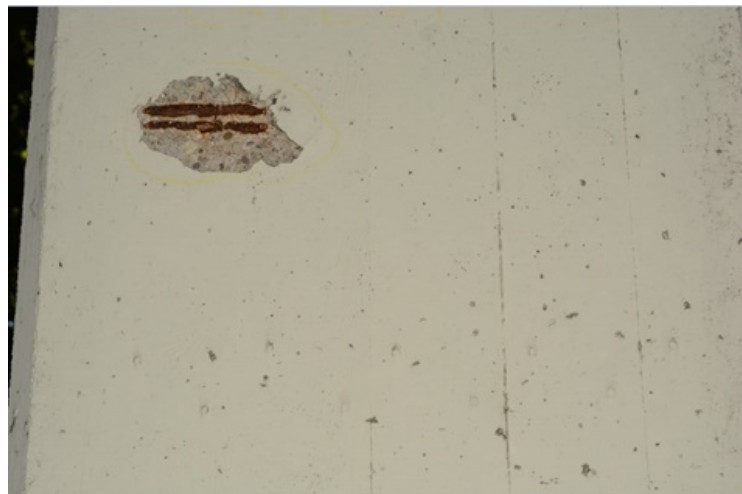

**Figure 7.** Image of concrete pathologies before detection with YOLOv4 [17].

Figure 8 shows the detection result of the trained network. A yellow rectangle, the bounding box, indicates the location of the identified pathologies. The network detected reinforcement exposure; in this case, a confidence of 70% detection is shown, fragmentation, and reinforcement corrosion.

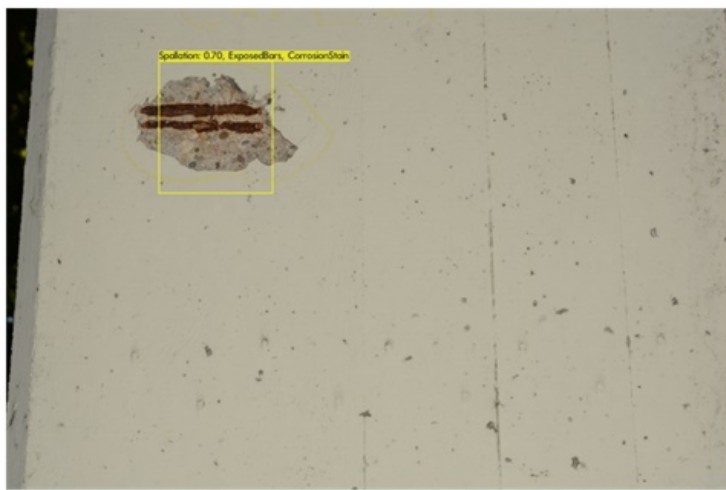

**Figure 8.** Image of concrete pathologies after detection with YOLOv4 [17].

### 3.2. Pathology Identification Using Cropped Images

The first experiment applies a structured convolutional neural network to recognize the two classes present in the dataset. Figure 9 presents examples of images from the dataset containing cracks and images without cracks.

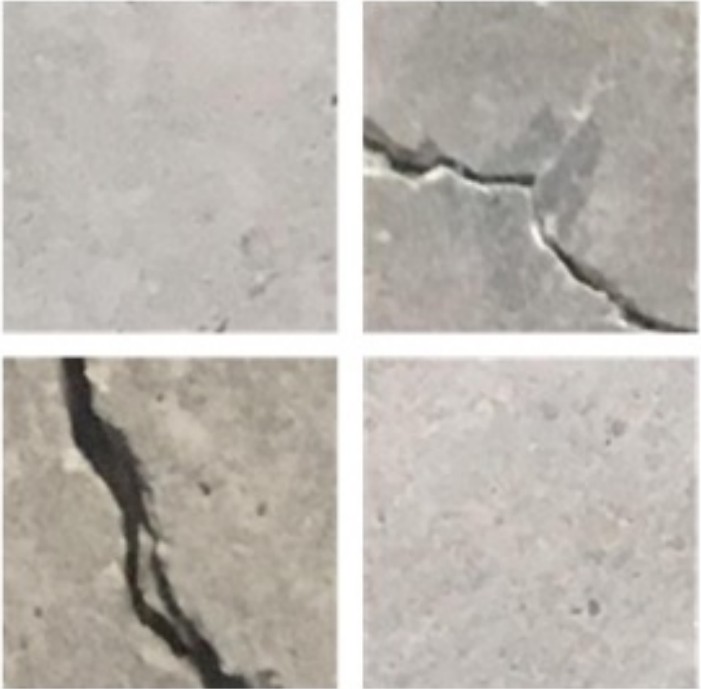

**Figure 9.** Images of the two types of classes in the dataset [18].

The input receives 227 × 227 pixels, three-channel RGB images. The network is fed with 20,000 images for each class, distributed as follows: 15,000 for training and 5000 for validation. Therefore, a total of 40,000 images is distributed evenly with and without cracks. This base is, therefore, balanced, containing the same number of images for the classes.

The first hidden layer comprises 32 3 × 3 filters. This layer makes use of the Relu activation function. Pooling follows this layer to reduce the dimensionality of the image. The next hidden layer comprises 64 3 × 3 filters. This layer makes use of the Relu activation function. Pooling follows this layer to reduce the dimensionality of the image. Following the same definition, except for the number of filters, the next hidden layer comprises 128 3-by-3 filters. Again, this layer is followed by pooling to reduce the dimensionality of the image. This layer also makes use of the Relu activation feature. After the previous sequence, the Flatten feature places the data in a single dimension. This layer is followed by a dense layer of a thousand neurons. Finally, the Softmax activation function and a layer of two neurons are used. The use of two neurons occurs because the final output is a classification for the two intended classes—images of concrete with and without cracks.

The training was performed with 50 epochs, aided by a GPU (Graphics Process Unit).

The results obtained were satisfactory, with an accuracy of 99.43% using cross-validation. Figure 10 shows the result of one section. Additionally, in this figure, it can be seen that the training and validation converged after about 20 epochs.

The training was performed with cross-validation. This technique prevents the choice of part of the base for training/validation from using a set that can vary greatly depending on the option. The dataset was divided into four training parts, leading to results of 99.45%, 99.43%, 99.41%, and 99.44%. This resulted in an average accuracy of 99.43% and a standard deviation of 0.017078.

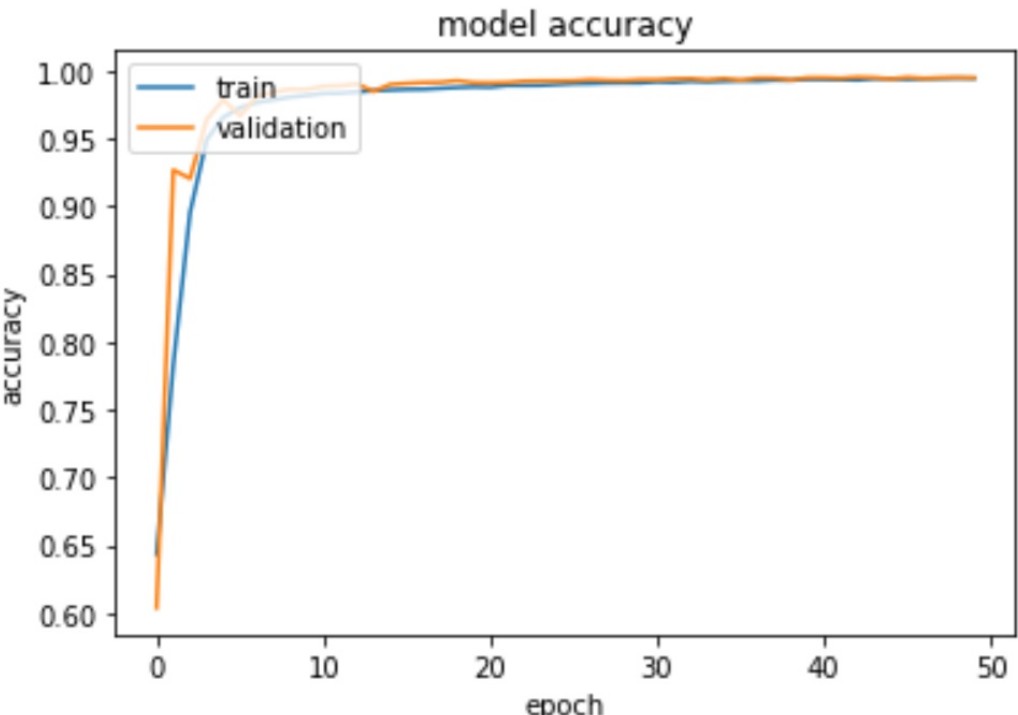

**Figure 10.** Accuracy obtained from CNN in one of the cross-validation steps.

Figure 11 below shows the evolution of losses during training in one of the cross-validation steps. It is also possible to see that the progress is satisfactory, being concluded with low loss.

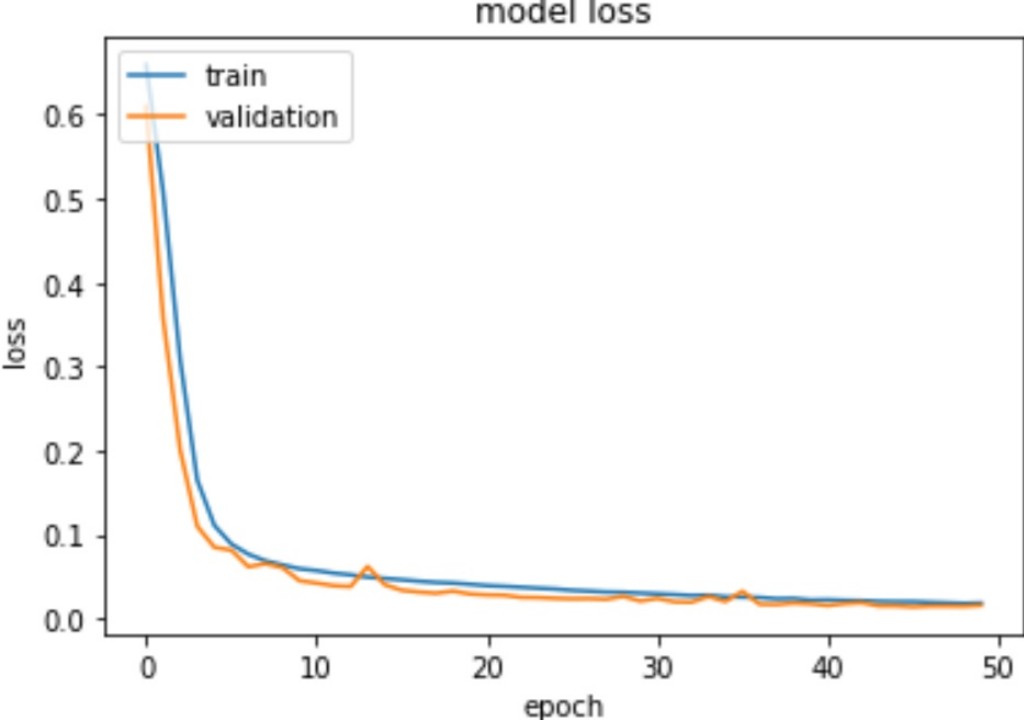

**Figure 11.** Loss obtained from CNN.

The literature presents similar work that can be used for comparison. Table 1 shows the results of other researchers so that there is a standard of comparison to indicate how good the result obtained is compared to different results.

The tests performed allow the performance of the network to be evaluated. Initially, a set of one thousand images was used to measure performance. Table 2 shows the confusion matrix, from which the 99.4799% performance of the neural network is inferred. The ROC parameter AUC is 0.999876. AUC corresponds to the area under the ROC curve, Figure 12, where zero is the lousy value and one is the optimal value.

**Table 1.** Comparison of the results of articles with similar experiments.

| Author | Accuracy (%) |
|---|---|
| Kim [8] | 99.98 |
| Ugne [25] | 99.00 |
| Proposed Method | 99.43 |
| Jitendra [9] | 98.60 |
| Alipour [7] | 98.60 |
| Pal [6] | 98.00 |
| Bai [4] | 95.92 |

**Table 2.** Confusion matrix for 4999 positive cases and 4999 negative cases.

| | Positive | Negative |
|---|---|---|
| **Positive** | 4969 | 30 |
| **Negative** | 22 | 4977 |

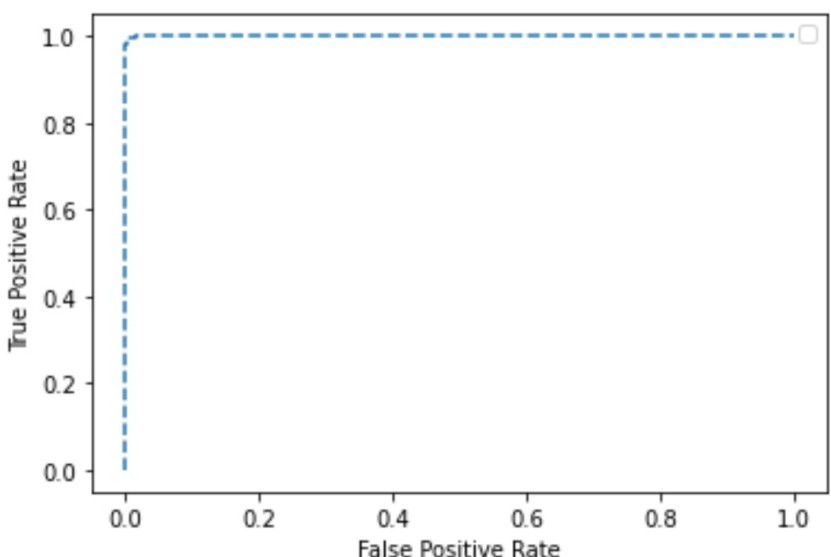

**Figure 12.** ROC AUC.

According to Figure 13, an additional test with two pictures—one of each class, one with and one without crack—allows to analyze the ability of the network to classify an image individually. This test aims to simulate the situation where an individual image needs to be classified and in the case of multiple images, the same feature can be used in a loop.

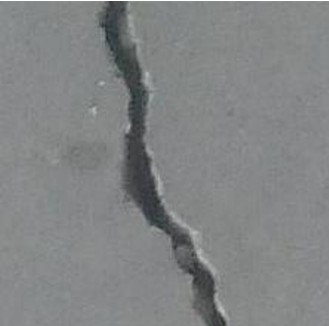

```
Positive test: This image is 100.00 percent Positive and 0.00 percent Negative.
```

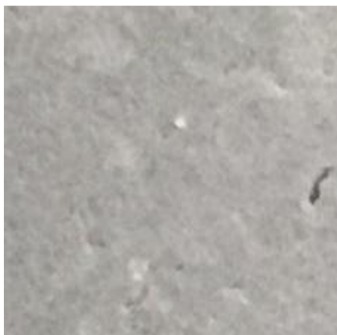

```
Negative test: This image is 0.00 percent Positive and 100.00 percent Negative.
```

**Figure 13.** Classification result for two images from the two classes of the [18] dataset.

## 4. Discussion

This paper presents a method for pathology detection and classification in concrete structures using deep neural networks. The method showed the possibility of speeding up the acquisition of results and obtaining more robust results, using deep learning knowledge for an area that Computer Science can enhance, Civil Construction.

Some datasets already enable the development of studies with concrete structures, showing that similar studies are already developing in this area. The results presented in this paper confirmed the possibility of gains in the studied area and showed that it is possible to develop resources with the existing material and techniques. Papers such as this one may stimulate the emergence of more datasets and, consequently, new applications.

The classification results were analyzed satisfactorily, with an accuracy and precision of 99.43%. The graphs show a convergence to good results, training, and validation, both in terms of accuracy and loss, which, in the loss, ends with acceptable low values. The confusion matrix also collaborates with the analysis to conclude that the neural network is suitable for identifying concrete pathologies.

It is possible to observe in the detection result that the method presents a good ability to detect pathologies in concrete, with individualized image testing in good conditions, but allows the questioning about improving accuracy. The trained network identified the proposed pathologies, but the accuracy needs to be better evaluated with more training epochs so that it can be concluded whether the obtained accuracy can increase with training for more epochs.

There are cases in which pathology detection, and consequent classification, can be difficult or even impossible because it is not well evidenced in the image. This is the case of stains that may overlap with the cases of "corrosion stains" and/or "efflorescence". Some cases of cracking, which have a straight geometric shape, can also be confused with structural joints or splits.

Since the detection system is not 100%, the processing may result in false positive regions, regions that have been indicated as containing pathology but do not have pathology. By associating the expert classification system with the sequence, it is possible to identify

these false positives. These regions do not have pathology and thus correct the false positive by disregarding it.

The qualifying practice sessions were better than the detections. Since detection involves the classification task, the difference in results must be investigated and tested under different conditions. The dataset used for the classification has 40,000 images. The dataset used for detection has only 1052 images. The difference is considerable, even considering that the images used for detection may have more than one pathology annotation per image, even considering a single pathology type.

As a suggestion, the dataset used for detection can be further investigated to identify a bounding box that has been poorly annotated. It is even possible to give more weight to annotations made by technical experts if necessary to increase the reliability of the analyses.

One point to consider for the detection result is the analysis of the annotated bounding boxes in the dataset and how well-fitted they are. The annotation work must be performed by a professional who has the ability to identify the pathologies, and, being a tedious and human activity, it may contain imperfections that impact the results.

Nevertheless, on the difference in performance between classification and detection, a point to analyze is the possibility of using the training performed in classification to improve the detection results since detection also has a classification step. One option is to use the classification training performed with a CNN to reduce the false positives obtained. CNN represents an essential solution for diagnostic imaging cases [26].

In practice, the method can automate identifying pathologies in concrete structures. The structure to be analyzed needs to be properly imaged, producing images of the highest possible quality. The generated images must be analyzed to verify their quality and, consequently, whether any image enhancement needs to be done. At this point of image enhancement, this method can be improved by defining automatic means for identifying which enhancements need to be performed and consequently applying the necessary enhancement. The enhanced images then proceed to detection, with the regions of possible pathology indicated in the structure-wide image. Classification is then performed on the regions separated by detection. The set of images identified and classified as pathologies and their respective classification can then proceed to the analysis by the expert, who will define the actions to be taken.

The proposed method exemplifies how it is possible to improve pathology identification tasks in concrete images, yet many improvements can be researched to achieve even more robust results.

**Author Contributions:** Conceptualization, A.C.d.P., J.d.C.N.D. and S.C.A.P.d.S.C.; methodology, J.d.C.N.D. and G.B.J.; validation, A.C.d.P. and S.C.A.P.d.S.C.; formal analysis, A.C.d.P.; investigation, G.B.J.; resources, J.d.C.N.D. and G.B.J.; writing—original draft preparation, J.d.C.N.D.; writing—review and editing, G.B.J. and J.d.C.N.D.; visualization, G.B.J., J.D.S.d.A., A.C.S. and A.M.T.d.S.C.; supervision, S.C.A.P.d.S.C.; project administration, A.C.d.P. All authors have read and agreed to the published version of the manuscript.

**Funding:** This work is financed by the FCT (Portuguese Foundation for Science and Technology) through the project UIDB/04082/2020 (CMADE).

**Institutional Review Board Statement:** Not applicable.

**Informed Consent Statement:** Not applicable.

**Data Availability Statement:** The data used are the dataset Ozgenel [18] and CODEBRIM [17].

**Acknowledgments:** This work was supported by National Funds through the Portuguese funding agency, FCT—Fundação para a Ciência e a Tecnologia, within project LA/P/0063/2020. The authors also acknowledge the Coordenação de Aperfeiçoamento de Pessoal de Nível Superior (CAPES), Brazil—Finance Code 001, Conselho Nacional de Desenvolvimento Científico e Tecnológico (CNPq), Brazil, and Fundação de Amparo à Pesquisa Desenvolvimento Científico e Tecnológico do Maranhão (FAPEMA) (Brazil) for the financial support.

**Conflicts of Interest:** The authors declare no conflict of interest.

## Abbreviations

The following abbreviations are used in this manuscript:

| | |
|---|---|
| MDPI | Multidisciplinary Digital Publishing Institute |
| CNN | Convolutional Neural Network |
| DB | Database |
| mAP | mean Average Precision |
| YOLO | You Only Live Once |

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
