# Peer review of "A Method for Detecting Pathologies in Concrete Structures Using Deep Neural Networks"

_applsci, doi:10.3390/app13095763_

Round 1

Reviewer 1 Report

Dear authors

The paper presents results in detecting the pathologies from different angle images containing the overall structure and also for the specific pathology  identification task for cropped images of the region of the pathology. 

Study is new and have potential to attract the readers of the field. Real time images validate the originality of the work. 

Script is well structured with proper connectivity in problem taken, findings and Results and discussions.

As per my point of view, this article have potential to be published in the mentioned journal of MDPI.

Thanks 

Author Response

Dear reviewer, thank you very much for your comments about our work.

Reviewer 2 Report

The paper presents a very interesting approach to the problem of detecting the pathology of reinforced concrete structures using deep neural networks. The paper can be published if the authors consider the following suggestions:

1.            Section 2.1 presents data on concrete pathologies. Please provide a more detailed report on the pathology of reinforced concrete with more literature references.

2.            In Table 1 we observe that the proposed methodology is slightly more efficient than other existing ones. Therefore, one could argue that the proposed methodology is not absolutely necessary. I would prefer that you add a paragraph presenting some arguments for the choice of your proposed methodology over those in Table 1.

Author Response

Dear reviewer, we greatly appreciate your contributions, which have undoubtedly improved the quality of the work.
The individual answers to each question are attached.

Reviewer 3 Report

This paper describes a method based on YOLO to detect five types of defects in concrete structures.

1.       The Introduction section lacks context on other useful applications of DL in concrete health monitoring, along with novel techniques used for the interpretation of DL results. Please improve this section by including the following relevant recent works:

a.       https://doi.org/10.1016/j.engfailanal.2023.107237

b.       https://doi.org/10.1016/j.autcon.2021.103936

c.       https://doi.org/10.1177/8755293019878137

2.       Honestly the presented methodology does not propose a new concept, there are several methodologies that follow the proposed scheme. What is the main contribution of the paper?

3.       In the dataset, how many images do authors have for each class? Did they perform labeling? More information is required

4.       Also the part of the image processing is poor. Please, enlarge this part

5.       Looking at the obtained results, is there a case in which the developed tool does not work?

6.       Overall results achieved are really poor in terms of mAP. The authors need to highlight how they interpret such results, and how they intend to improve and achieved them in future works.

7.       In the end, how the method could be used in the practice? How can it be improved?

8.       Overall, the paper can be considered for publication after a major revision. I also suggest double checking the English, the style, and the flow of the narrative.

I also double checking the English, the style, and the flow of the narrative.

Author Response

Dear reviewer, your contributions undoubtedly improve the quality of the submitted work. We are very grateful for your comments.
The individual answers to each question are attached.

Round 2

Reviewer 3 Report

The authors have successfully answered to all issues. Hence, the paper can be considered for publication.